# In WAZE we trust? GPS-based navigation application users' behavior and patterns of dependency

Tal Laor[1], Yair Galily[2]*

**1** Founder & Head of Radio, Broadcast and Content Production Studies Track, School of Communications, Ariel University, Ari'el, Israel, **2** Head, Sport, Media & Society Research Lab, Sammy Ofer School of Communications, Reichman University, Herzliya, Israel

☯ These authors contributed equally to this work.
* ygalily@idc.ac.il

**Data Availability Statement:** All relevant data are within the paper.

**Funding:** The author(s) received no specific funding for this work.

## Abstract

Functional technological applications have become an integral part of our lives changing our patterns of reasoning and behavior. The current study examines whether, how and why use of WAZE app, a popular GPS-based navigation application, demonstrate behaviors and patterns which resemble those of technological dependency. We conducted semi-structured in-depth interviews with 50 WAZE users. The questions took inspiration from the model of IT addiction, which identifies six behavioral parameters: withdrawal, conflict, mood modification, relapse, tolerance, and saliency. The novelty of the study lies in the evidence of patterns and behaviors which resemble technological dependency on the WAZE app. The findings indicate that WAZE app satisfies users' needs driven by functionality. Four behavioral characteristics associated with IT addiction are applicable to WAZE users: mood modification, conflict, relapse, and withdrawal. The study concludes that functional technological applications may trigger behavioral indicators of technological addiction.

## Introduction

Addiction is a state in which an individual develops a dependency on a specific object or behavior. Information technology (IT) addiction [1] is a specific type of behavioral addiction that occurs when an individual is in a state of loss of control vis-à-vis a technological resource [1, 2], and is defined as a psychological state of unconscious dependency on technological use.

The current study focuses on the use of WAZE, a satellite-based navigation application that provides real-time information on users and their driving routes based on the location of their mobile device. Compared to other GPS devices, WAZE is unique because of its social network, which operates as a circle of friends and a platform for message exchange, and because it learns about its users' driving in order to offer driving routes and real-time traffic updates [3]. Motivation studies found that the main reasons for using WAZE are simplicity, user friendliness, route customization, avoidance of traffic jams, and the option of creating routes with stops on the way [4].

**Competing interests:** The authors have declared that no competing interests exist.

WAZE was selected as a study case because functional technological applications have become an integral part of our lives changing our patterns of reasoning and behavior; specifically, WAZE has become an integral part of the everyday driving experience. However, we found few studies on the application itself and its usage patterns. Moreover, studies on dependency on WAZE are rare, which underscores the significance of the current study and its novelty and contribution to scholarship and society. The novelty of the study lies in the examination of the experiences of WAZE users to explore whether how and why the app meets the needs of the users, if the excessive use of this functional technological app demonstrates behavioral characteristics that resemble those of technological dependency. To that end, we conducted semi-structured in-depth interviews with 50 WAZE users. The questions were inspired by the model of IT addiction [1], which identifies six behavioral parameters: withdrawal, conflict, mood modification, relapse, tolerance, and saliency.

## Theoretical background

Dependency is the quality or state of being influenced or determined by or subject to another [5]. When an addict (defined above) interacts with an object for which they feel a strong need, the brain releases dopamine, a substance similar to adrenaline, which immediately causes a sense of relief, excitement, and elevated mood. When the addict is separated from that object, a sense of emptiness and depression arise and an urgent urge to satisfy the need in question. Addiction involves mental, physical, and social factors. Individuals may become addicted if they develop a dependency on a chemical substance, but they may also become dependent on certain behaviors, the use of certain devices, etc. [6].

According to Marlatt, G. Alan, et al. [7], behavioral addiction is based on recurring behaviors and habits that are associated with personal and social problems that may be subjectively experienced by the individual as a sense of loss of control.

The essential feature of behavioral addictions is the failure to resist an impulse, drive, or temptation to perform an act that is harmful to the person or to others. The repetitive engagement in these behaviors ultimately interferes with functioning in other domains. Behavioral addictions are often preceded by feelings of "tension or arousal before committing the act" and "pleasure, gratification, or relief at the time of committing the act". Many people with behavioral addictions report an urge or craving state prior to initiating the behavior, as do individuals with substance use disorders prior to substance use.

Additionally, these behaviors often decrease anxiety and result in a positive mood state or "high" [8, 9].

Behavioral addiction that involves an individual's interactions with a device is known as IT addiction [2]. IT addiction is a specific type of behavioral addiction that occurs when an individual is in a state of loss of control regarding a technological resource [1, 2], and is defined as a psychological state of unconscious dependency on technological use. The typical features of behavioral dependency are: (a) salience–the technology dominates a user's thoughts and behaviors; (b) withdrawal–negative emotions arise if the individual cannot use the technology; (c) conflict–the use of the technology conflicts with other tasks, which impairs normal functioning; (d) relapse–the user is unable to voluntarily reduce their use of the technology; (e) tolerance–increasingly large doses of activity are needed to achieve the same effect; and (f) mood modification–use of the technology offers thrill and relief, and modifies mood.

Internet addiction is not only a function of one's frequency of Internet use, but is also related to the one's surfing environment and its social norms, which influences individual behaviors. Consequently, it is difficult to precisely define "normal" Internet use, or the point at which use and deviation can be identified [6].

## Internet addiction

In the past, access to the Internet was limited to the government and specific elite groups such as scientists, engineers, and mathematicians. Today, the Internet is accessible by the general public and is used widely by an enormous number of individuals. Statistics point to rapid growth in the number of mobile devices in individual hands and accelerated growth in the use of text messages that obviate the need for voice calls and reduce costs of communication [10]. The use of technology has become excessive, to a point that significantly impairs users' lives when users develop a dependency on text messages in everyday life. In Japan, for example, women who leave their mobile phone at home tended to be more irritated [11]. Furthermore, people with social anxiety and a tendency toward impulsive behavior are likely to be more dependent on the Internet and prefer to maintain their relationships with others online. As with other behavioral addictions, Internet abuse has been a controversial topic; one of the most challenging tasks has been to arrive at a comprehensive definition of the concept.

Terms such as Internet addiction, obsessive Internet use, problematic Internet use, obsessive computer use, computer addiction, and pathological Internet use represent obsessive and excessive use of the Internet, and mood modification that occurs when access is denied or the technology is unavailable [5, 12, 13]. Internet addiction may impair the addict's functioning. For example, students may be distracted by surfing the web when they study and people may also use the Internet as an excuse for avoiding everyday tasks [14].

The term *pathological Internet addiction* was first coined in 1998 to describe Internet addiction as an impulse control disorder that does not involve an intoxicant [15]. The diagnostic criteria for Internet addiction are similar to those used for substance dependencies. However, Internet addiction is not included in the DSM-5 [16].

## Mobile telephone dependency and addiction

The primary motives of mobile phone use are information, entertainment, new relationships, security, and peace of mind [17]. Several terms are used to describe heavy mobile phone use, including problematic use of mobile phones [18–20], dependency [21], dysfunctional use [22], and addiction [23]. In the clinical psychiatric literature, mobile phone addiction is defined as a behavioral addiction that includes the parameters of withdrawal, longing, and loss of control. Although there are differences between ordinary addictions and mobile phone addictions, they share several parameters with other types of addiction, such as addiction to sex, alcohol, the Internet, and online gambling [19, 24].

Mobile phone addicts are also likely to be in a state of emotional instability [25]. The literature indicates that loneliness and social anxiety are positively associated with smartphone addiction, and lonely individuals tend to rely more heavily on smartphone-based communications than on face-to-face communications to relieve their loneliness, which increases the probability of post-traumatic stress disorder (PTSD) and reduces their perceived social support [4].

However, it is also possible to point to positive outcomes. For some individuals, the Internet functions as a bridge between online networked life and offline life. Internet and smartphone use are coping mechanisms to compensate for psycho-social problems, satisfy needs, cope with negative past experiences, and increase individual well-being [26]. It has not yet been determined whether technological devices can be seen as a cause of addiction. The definition, diagnosis, and treatment of IT addiction are still in their infancy, and research has yet to resolve these issues.

## The WAZE application

WAZE is one of the most popular applications, especially in the category of GPS-based applications, and facilitates information-sharing on traffic conditions on the ground in real time. In WAZE, user-posted information about the traffic is constantly evaluated by other participants [3]. The GPS-based application generates real-time information on users and their routes based on the location of users' mobile phone. Compared to traditional GPS-based devices, WAZE has unique features that transform its users into a social network, such as a circle of friends and platform for message exchange. WAZE constantly learns its users' habits in order to suggest driving routes and provide real-time traffic updates.

The application allows users to report accidents, traffic jams, road works, hazards, etc. All of the alerts posted by users are subject to other users' evaluation; users tap the "like" button to confirm the alert, or the "deny" button if they feel it is not accurate.

Of the few studies to have been conducted on WAZE, several explore the use of technologies to promote road safety [27], drivers' motivation to share their information with other WAZE users [3], and social aspects of use, such as voluntary assistance in mapping the roads for other drivers [28].

Motivation studies found that the main reasons for using WAZE are simplicity, user friendliness, route customization, avoidance of traffic jams, and the option of creating routes with stops on the way. The majority of non-users of WAZE were likely to adopt the application after exposure to its benefits [4]. The interface is also similar to a gaming interface that heightens consumers' engagement. At the same time, these features generate concerns that use of the application promotes maladaptive driving patterns and creates distractions for drivers [28].

## Location-Based Services (LBS)

In recent years, the use of location-based services (LBS) has grown as various technologies have increasingly adopted a layer of information based on users' location. Examples include the "check-in" option on social media posts, navigation applications such as WAZE, location-based games, and location-based information embedded in images [29]. Today, 74 percent of smartphone users use navigation apps and 30 percent use social media that include their location in their posts [30].

Location-based technologies expand users' abilities in various ways. They offer information on points of interest and entertainment options in AR games [31], and promote connections with friends [32] by updating users' presentation of the self on social media that corresponds to users' locations ("showing off") [33]. Scholars have voiced concerns about how location-based technology, combined with social media, infringes upon, or has the potential to seriously infringe upon, users' privacy [34, 35].

## Research questions

Our research questions seek to find whether, how, and why the WAZE app meets the needs of its users, and whether excessive use of the app demonstrates behavioral characteristics that resemble those of technological dependency.

## Methodology

Semi-structured in-depth interviews were conducted with 50 WAZE users, between the ages of 23 and 55, including 25 older drivers who had previous driving experience using maps, without the use of the application, and 25 younger drivers whose driving experience had always included the application. Half of the interviewees of each group were females. All

interviewees were experienced drivers and had held a driver's license for at least five years, had WAZE installed, and typically drive at least three times a week. 3 research assistants were asked to reach 20 potential interviewees who meet the up mentioned criteria. That is, each assistant reached 10 males and 10 females. Each gender is consisted of the older group and of the younger group equally. Finally, 50 interviewees of that pool were selected.

The interviews varied in duration from 47 to 100 minutes ($M_{time}$ = 71.75, $SD$ = 17.35 minutes). In line with the interpretivist paradigm, the interviewer wrote detailed analytic memos throughout the interview, elaborating on key issues and examining meanings. The interviews were audio recorded and the data were then transcribed verbatim, mainly by the second author. This resulted in 284 double-spaced pages of transcribed data.

To identify the behavioral characteristics in WAZE usage, interview questions focused on interviewees' consumption patterns of the application, the meanings of their use experience, and their attitudes and interpretations of WAZE. The interviewers conducted an open conversation and allowed interviewees to speak freely about any topic they felt was relevant to the topic of the study. Interviews took place in face-to-face meetings with interviewees in order to establish conditions that foster deep, candid conversations. Interviewees were informed at the outset that their interviews would be used exclusively for academic purposes and that their names would be omitted from the report of findings.

The questions took inspiration from the literature review regarding technological dependency and the model of IT addiction [1], which identifies six behavioral parameters: salience, withdrawal, conflict, relapse, tolerance, and mood modification.

Several significant questions were involved in the interviews, as follows:

Which needs does the application satisfy for you? For what type of drives do you use WAZE?

How safe do you feel on the road? How strong is your sense of orientation on the road? Does using the application diminish your sense of orientation on the road?

If you had to leave only three apps on your phone, which ones would you leave?

What feelings and emotions does the use of the application evoke in you? Have you ever found yourself unable to use WAZE? How did you feel in those situations of malfunctions in the app? When the application is working, how does it make you feel? What emotions are evoked while driving with WAZE? Has the application changed your driving experience? Have you ever chosen to drive without the application?

## Data analysis

The Atlas TI software was used to organize the generated data [36]. Thematic analysis adequate for this subject and dataset was implemented (Braun et al., 2016). This analysis involved several phases: (1–2) familiarization and coding, (3–5) theme development, refinement and naming, and (6) writing up [37]. Initially, all interview transcripts and interview notes were read several times. The first author attempted to adopt an indwelling posture by going through the data, immersing in it, and empathetically understanding the drivers' point of view. Data were analyzed both inductively and deductively [38], as the researcher was moving back and forth between the drivers' personal experiences and the conceptual lens. An inductive approach was first assumed, and the researchers attempted to identify meaningful data units (i.e., raw quotes were coded) related to the interviewees driving experiences and categorized them into sub-themes (i.e., code groups).

## Ethics

Institutional ethical approval was secured from the second author's university. Participation in this study was on a voluntary basis only. At the beginning of each interview, the interviewer

explained the nature and purpose of the study and asked the participants to sign a consent form. The participants could withdraw from the study at any point and could ask that any data would not be included in the data analysis or presentation. Data collection ensured data security and the protection of participants' confidentiality, as much as possible.

## Findings

### Use for personal well-being: Meeting needs

Neto et al. [3] found that using WAZE has become a natural, embedded interaction, an unconscious technology rather than a forced action related to traditional forms of work. In the current study, all interviewees stated that WAZE was one of the three applications that they would not delete. In order to prevent social desirability bias, this question was posed at the beginning of the interviews, before the interviewer posed questions that focused specifically on WAZE.

> S (female aged 25) explained, "These are the applications that I use the most, on a daily basis."

> B (male, aged 54) added, "Those are my most useful [applications] during the day. The things I use the most and am most connected to."

Interviewees defined the app as a fundamental everyday driving aid rather than a supplementary aspect of their driving experience. In other words, denying them access to the application would be experienced as the denial of a basic need and would adversely affect their everyday functioning. Half of the interviewees said that WAZE is an application that they "can't live without."

First and foremost, WAZE satisfies interviewees' needs, which explains its wide use. The majority of interviewees stated that they use the application to facilitate driving, as it shortens driving time by avoiding traffic, prevents them from being stopped by the police or caught by traffic cameras, and allows them to schedule their trip according to the conditions on the road:

> If there's an accident or a road is blocked or there's traffic somewhere specific, it helps to know about the police and traffic cameras on the way and to avoid tickets. (D, female, aged 26)

> [It offers] important information that allows you to organize your schedule and receive advance warnings and know what's going to happen. Especially police and traffic cameras on the road, and warnings of roadblocks, and knowing your arrival time. (T, male, aged 62)

> I had an interview in the north. WAZE warned us that we would be a little late so we could let the interviewer know ahead of time that we would be late. (Y, male, aged 25)

These findings shed new light on motivations of use and are consistent with the benefits of using WAZE identified by Noerkaisar et al. [4].

The majority of interviewees noted that they also use the application during short, routine drives for information on the traffic jams, arrival times, road hazards, etc. Moreover, the interviews clearly indicated that WAZE meets users' need to maximize their use of time, efficiency, and effectiveness in their everyday lives, as the following quote illustrates:

> It simply creates order in my life, not only on the routes that I know, [it] even [lets me] know how much time it takes to get to work. It's important for me to know what time I will arrive and exactly what road to take. I take many different routes to work and it helps me

see which is best. It helps me to be able to tell people my arrival time. It helps me be professional because I can schedule meetings accordingly because I know exactly what time I will reach each and every point. (D, female, aged 45)

The next interviewee similarly recounted the advantages that the application offers him:

The fact that WAZE can tell me how long it will take me and exactly when I will arrive, and at the same time calculates the best time for me to leave . . . We didn't grow up with something like that. It also helps me in my work: When I have a meeting, it's easy to coordinate when we all arrive. WAZE also tells me the best and most efficient route to take. It saves me time, money for gasoline, it saves arguments in the car over who is navigating and when we should ask directions from people on the street . . . (B, male, aged 54)

Another interviewee recounted a situation in which he was unable to use WAZE because of his mobile phone battery: "I wasted so much time that I looked around for somewhere to plug in and charge my mobile phone . . ." (G, male, 36)

We can see that interviewees emphasize the efficiencies generated by the use of the application, for planning their day, getting organized, and reducing hindrances such as traffic jams and waiting for others. One of the older interviewees summarized her own experience with WAZE by saying, "It's a different kind of driving experience." She was amazed that "once, we never knew when we would arrive. Now, everything is updated online . . ." (M, female, 56).

WAZE is especially suited for the current generation, in which knowledge is recognized as power, and people feel the need to be constantly in control, and therefore accurately satisfies users' needs. This argument is supported by the finding related to sharing. WAZE is used by drivers sharing functional information about traffic and road conditions, such as by reporting road accidents, hazards, police, etc. [28], and the majority of interviewees (43) stated that they do not use the application to share other information at all. Reasons for this ranged from a lack of interest in sharing to a fear of getting a ticket for using their mobile phone while driving. When asked, they stated that they felt no obligation to share information on the application. These findings indicate that users are discerning users of the application who select uses according to their own needs and convenience.

In summary, despite the existing literature on the application's sharing features and the community of drivers that it creates [3, 39], our study found that interviewees do not typically use the application to share information, and instead use it exclusively for their personal benefit, to ease their driving experience, bring them to their destination efficiently, quickly, and with maximum safety, saving them potential physical and monetary losses. Interviewees' responses indicate similar patterns of use. In addition to using the application to guide them to their destination, many interviewees use additional application features that make a significant contribution to their driving experience (alerts about police, traffic cameras, traffic jams, etc.), and specifically assist them in planning their schedule. These findings indicate that WAZE satisfies users' needs efficiently and effectively, and maximizes their time efficiency.

## Dependency on the application: Traces of relapse

WAZE is more than a mere driving assistant; it has become an integral element of the practice of driving itself. The ability to open, close, and confirm roads on a map interface already existed, for example, in state agencies' databases of satellite-based navigation companies [4]. This significant transformation is reflected in the interviewees' responses. Interviewees can be divided into two groups based on their response regarding their dependency on the

application. Drivers who had previous experience driving without the app stated that they felt no dependency on the application, while younger drivers, for whom WAZE was the only significant driving assistant that they know, felt that their driving depends on the application. Out of the 25 younger interviewees, 15 felt that their driving was dependent on the application and that the application controlled their behaviors. They were reluctant to leave their house without it and felt unable to manage without it if the route was unfamiliar to them. Y (male, 25) explained his dependency on the application in the following words: "I rely on WAZE too much. I am dependent on it because it gives me the confidence to go from one place to another. I feel less confident if it suddenly stops working. I would be very fearful of driving without WAZE to an unfamiliar place."

Similarly, R (female, 25) stated that she "won't leave the house without WAZE, I really need it in my life." Another interviewee said, "I can't manage without it; it's with me wherever I go. . ." (M, female 56). Yet another said, "I really need WAZE to drive. It's really difficult for me without out. Especially if I'm going somewhere and I don't know the way. . ." (D, female, 45). Another interviewee is so dependent on WAZE that she even uses the application when she walks. "I need WAZE. I use the application even when I walk, to know the directions. . ." (L, female, 26).

The remaining 10 younger interviewees stated that while they do not feel that their driving depends on WAZE, not using the application would make their driving significantly more difficult. M (female, 27) stated, "I think that I wouldn't be dependent on it but it's very significant. I mean that I'll go even if I don't have a battery, but when it's there it saves a lot of headaches, [you don't have to] stop on the way to ask, look around, drive in fear and check the signs."

The same interviewee noted that she does feel dependent on WAZE on roads that she considers to be more dangerous: "I have this fear [of those roads] and so WAZE gives me confidence . . . [I know that] it's silly and its psychological, but I have this thing where if I am afraid of driving there, and I drive a lot, then WAZE tells me the way and it calms me . . . it's like suddenly there's someone there talking to you" (M, female, 27). On those occasions, she stated, she would not drive without WAZE, which suggests that even drivers who reported they are not dependent on the application may still have some degree of dependency.

In contrast to the younger group of interviews, the majority (20) of older interviewees stated that they would not be reluctant to leave the house without WAZE. One male (age 55) explained: "It's a matter of convenience. I would drive and manage. [After all,] What did we do before [we had] the application? I had a book of maps in the car and would check it in advance. I learned the route in advance and I asked [people for directions]. And it worked well."

Previous experience constitutes a significant reason why interviewees felt that they would manage as they had in the past. S (female, 50) stated, "If I couldn't use [the application] then I would print out maps . . .like what we did in the past, we used to access "maps" through Ynet or something similar. I don't feel that there's any dependency." Another interviewee noted similarly that "I would prefer not to drive without it but if I had no choice, I would get [the route] from the Internet in advance, print it or something, like I used to do in the past, and use that. . .." (D, male, 44).

The interviews indicate that the differences between the younger and older interviewee groups stem from certain skills and competencies that the younger application users never needed or developed, such as navigation skills, road memory, spatial attention, and map reading. Older application users noted that a long time had elapsed since they last used these skills and, in some cases, the use of the application diminished these capabilities, but they felt that they would be able to rely on them if necessary, as the following quotes illustrate:

WAZE has made me more fixed in my habits. You don't really try to remember to turn right at the next turn, you rely on it, and it takes some of the responsibility from you. If you use same route again and again, eventually it would sink in, but ultimately, even we have WAZE so you don't even try . . . (L, female, 27).

Another interviewee offered the following description:

It changed [my] map skills for the worse . . . in the end you are dependent on the application and you remember fewer names and things stick less in your memory after you start using the application . . . (M, female, 56)

The following interviewee similarly described how WAZE makes him feel as if he "sheds responsibility" and supposedly can drive automatically without paying attention to the road:

It makes me look at the environment less because you're very safe, you tell yourself that it's alright because WAZE is telling you when to turn, so you don't really concentrate on the road . . . you drive blind because WAZE is telling you . . . (R, female)

And yet another interviewee stated:

"I guess that if I would use the application less, I would know the roads I drive on better . . . I mean, it's a problem, because I am less familiar with the roads because of the application . . ."

It is possible that, as a result of their under-developed skills mentioned above, younger drivers did not develop the important ability to communicate with people they do not know; this was reflected in the amazement expressed by a young interviewee in response to the question of whether she would leave the house without WAZE:

". . . I wouldn't leave the house. Do you think I would ask people on the street how to get there?! Unless it's someone I really know well, someone from home or work, then maybe. For anything else I would never drive. I can't. . .. (L, female 27)

We can conclude from the interviewees' responses that their behavior is affected strongly by the application, and this is especially true of the younger interviewees. According to the application, one determines whether he or she will leave the house or not, or go to work, if they fear traveling on unfamiliar roads. These findings may be considered in line with the findings of Infoplant [11], who argued that technology use has become excessive and uncontrollable, to the extent that it impairs users' lives. Harm is reflected in developing a dependency on WAZE and a fear of leaving the home without it. Even the older interviewees, who denied that they were dependent on WAZE due to their past experience, noted that the application adds convenience to their life, which implies a relationship of dependence [26].

## Anxiety as a response to malfunctions in the application: Traces of withdrawal

Problematic use of technological devices leads to mood modification when the consumer avoids their use [40]. More than half (36) of the interviewees reported a sense of anxiety, stress, and pressure when their application stops working. D (female, 26) recounted how the GPS

services in her mobile phone stopped working during a trip. "I found myself in real trouble, because I was so dependent on WAZE that I thought that I lost it." Another interviewee shared that "my heart stopped beating when my WAZE suddenly shut down and stopped working" (L, female 27).

The interviewees explained that they felt anxious because they did not know where they were or how to proceed. The following story illustrates this feeling:

> During a trip in the summer, my phone overheated and just turned off. I tried to stop on the side of the road and cool it off with the air conditioner, but it wouldn't turn on until it really cooled off. I started to feel stressed out, I felt my blood pressure rising. If it had been a place that I know, it wouldn't be too terrible not to know where the traffic is congested, but if it's a strange place and I don't know the way at all, then it's really stressful. There's even a sense of vulnerability, a state of helplessness, however pathetic that sounds. You don't know how you'll get back on the right road and reach your destination..." (D, female, 45).

Many interviewees (20) similarly described a sense of helplessness on occasions when WAZE was not operating: "I feel helpless. I feel that someone took a bone from my body. I don't know what to do–turn left, right, or go straight" (A, male, 26).

Another negative emotion that arose in such occasions was frustration. Interviewees stated that if the application stops working for one reason or another, they would feel frustrated due to their dependency on the application.

> I would feel frustrated because I'm dependent on it. Without it, I don't know where I need to go, in what direction. (S, female, 50)

Another interviewee described a situation in which her application stopped working and the effect that it had on her:

> I felt pressure. I was dependent on the application to get there and I became stressed because I kind of developed a dependency on it. [There was a sense of] uncertainty, a loss of control over the situation. (R, female, 43)

A sense of loss of control was also reflected in another interviewee's statement:

> "Loss of control. I just can't find my way without it." (A, female, 25)

A (male, 54) added, "It felt a little like helplessness to me and it was even irritating. Because the moment you turn the application on, you say to yourself in your head, 'alright, now it will do all the work,' and you rely on it. The moment that [the responsibility] goes into your own hands, and especially when you can't even fiddle with it because you're driving, then it's aggravating."

The findings of the current study indicate that negative emotions are generated when users are unable to use the application. The application has mental implications related to anxiety, which resemble the effects found in other types of technology use [4], and a sense of helplessness that results from users' dependency. WAZE has become the source of satisfying a need so basic and mundane that, when it is impossible to use the application for various reasons such as a malfunction or dead battery, the situation is described almost as being like lacking air for breathing and as a disruption to one's daily schedule, and leads to frustration and other negative emotions.

### The application's effect on users' road orientation: Traces of conflict

The findings of the current study also indicate that use of the application comes at the expense of familiarity with roads (in other words, use of WAZE comes at the expense of other important activities Giles et al., 2011). Participants in the current study were asked whether using the application diminishes their sense of orientation on the road during their normal everyday functioning and driving on the road. Half of interviewees (25) recounted that when they use the application, they do not notice the road; they feel as if they are driving "without seeing" and they shift the responsibility for driving from themselves to the application. As a result, the use of WAZE app impairs normal functioning of taking responsibility of the driving [11, 14].

You rely it and that removes your responsibility. (D, female, 51)

Another interviewee similarly explained:

It's like you hand over your responsibility for the road; you are told what to do and you stop thinking. It's like I drive on auto-pilot. Your head is preoccupied with other things and then you think less and use your judgment less. (S, female 50)

Y (male, 25) described his feelings as follows:

When I drive, I don't really take a careful look at the signs, [in any case] WAZE might give me other instructions and so, in effect, [the signs] don't matter to me, I just listen to the instructions. Whatever WAZE says, I listen and rely on it. If WAZE tells me not to take a specific road, I won't take it.

R (female, 25) described the situation as follows:

I have no idea where I am going. WAZE says "right" so I turn right. I have no idea about the various signs. They might be directing me to the ends of the earth and I wouldn't know it.

The interviewees spoke of their sense of being distracted from driving due to the attention they direct to the application, which is consistent with findings of studies on the distractive effect that the Internet can have on people while they are studying (Young, 1998). The following quote illustrates this distraction effect:

You don't really concentrate on the road. If you don't have WAZE, you rely on the signs or other things like that, then you concentrate on the road more, on what really is going on around you, on the signs, and turns . . . When WAZE is on, we pretty much drive "blind" because WAZE tells [us what to do]. (H, female, 27)

Another interviewee stated:

You are so careful not to miss a turn that you look at your mobile phone and not at the road. (G, male, 36)

In summary, these findings indicate that WAZE serves as a more reliable and convenient source of information during driving than the driver themselves. In other words, use of WAZE diminishes users' skills and competencies by eliminating the need for them.

Application users devolve their responsibility for skills such as road memory and spatial orientation to the application. As a result, the application has potentially significant effects on various functions and other basic actions by obviating users' need for them. This finding is in line with those of earlier studies on digital addiction [11, 14].

## From negative emotions to positive emotions: Traces of mood modification

The majority of interviewees (35) in the current study answered in the affirmative when they were asked whether they feel relieved when the application is in operation. The following quote is illustrative:

[When WAZE is on] I don't think twice before driving. I have no problem driving to Eilat or to Kiryat Shmoneh because I know I have WAZE. It gives me a sense of calm. (Y, male, 27)

Moreover, 16 interviewees reported that they use the application because it gives them a sense of safety on the road while driving:

"I feel safer on roads that I'm not familiar with." (A, female, 25)

Another interviewee described her sense of calm when the application is in operation:

I feel as if everything is going as it should. A sense of security and serenity, I know that I will manage on my own and can reach any destination properly. (D, female, 51)

Another interviewee stated:

It generates a sense of security. I feel as if I can control my driving with no problem, that I will be able to reach my destination in the quickest and most efficient way. I am calm and feel that everything is going as it should. (Y, male, 25)

Another interviewee stated that he felt a sense of peace when the application was working in the background because "I know that I am in good hands. . ." (M, male, 28). Another interviewee explained this sense of calm as the ability to reduce his typical degree of vigilance when WAZE is not in use. He stated, "Usually, even if you turn on the voice alerts, it has a calming effect. You can do other things. You're not stressed or constantly vigilant about where you have to go, and you can relax. . ." (D, male, 35). Another interviewee explained that WAZE allows her to relax, knowing that even if she makes a mistake, the application will correct her and guide her to the right place:

It gives a lot of confidence to a person on the road because it relaxes you when you know that even if you make a mistake, there's someone there to calculate your drive, which is important and helpful. (L, female, 26)

Several interviewees in the older group noted that their sense of peace and calm is related to the fact that using WAZE eliminated frequent arguments about driving directions or the need to stop and ask for directions.

According to these findings, WAZE use affects users' mood; this is consistent with previous research showing that increased use of a mobile phone may lead to emotional instability [25]. Driving while using WAZE creates a sense of peace and calm on the road. Interviewees

reported a sense of relief because they experience a sense of safety when the application is operation, and a sense of calm confidence that the drive will proceed without issue because the application knows the best and quickest route to the destination. Users rely on the application and therefore experience negative emotions when the application is not in operation, and positive emotions when it is properly operating.

## Discussion

The Internet and smartphones are now accessible almost anywhere and at any time, and an increasing number of functional applications have entered our lives to satisfy diverse needs, becoming part of our daily routine. Over the past decade, an increasing number of devices have been developed to assist drivers in achieving these goals. WAZE is a popular functional navigation app that is based on social big data [41], is updated in real time by its users, and conveys these updates to users in real time [3, 30], to generate the shortest route for each individual driver. In general, WAZE meets drivers' needs, as the purpose of driving is to reach destinations in the safest and shortest way possible.

The current study has examined whether, how, and why use of WAZE demonstrates behaviors and patterns that resemble those of technological dependency.

Fifty individual semi-structured in-depth interviews were conducted with WAZE users. The findings indicate behaviors and patterns that resemble some degree of technological dependency. According to the literature, technological addiction has six features: withdrawal, conflict, mood modification, relapse, tolerance, and salience [1]. The interviewees in the current study reported behavioral characteristics, when using WAZE, which may show resemblance to four of the behavioral characteristics of IT addiction as explained below.

Interviewees describe a change in their mood when WAZE is accessible. They feel calm, safe, and less vigilant. That is to say, the application creates a broad safety net for all their behaviors on the road and allows drivers to focus exclusively on the technical and safety aspects of driving. Users feel that someone else is taking care of them, offering them the quickest and safest route to their destination. Even when road conditions change, the application informs them of the change in real time (via the "recalculating route" feature). On the other hand, when users are unable to use the application, they experience a sense of withdrawal, which is reflected in negative emotions they reported, such as helplessness, uncertainty, anxiety, stress, frustration, and fear. Thus, WAZE use can generate a sense of emotional instability, which may resemble mobile addiction [25]. Apparently, frequent WAZE use leads to a conflict with other activities by shifting the responsibility for driving from the driver to the app. As a result, senses and competencies such as orientation, memory (of roads and routes), concentration, road familiarity, time estimation and time management are diminished, which echoes the findings of studies on digital addiction [14]. It seems that drivers who use the application lose skills and competences that are critical for navigating, such as orientation. These may be considered consistent with the findings of studies on behavioral addiction, which is characterized by recurring behaviors and habits that are potentially experienced as conflict and as a sense of loss of control [11, 14] which leads to relapse and diminishing skills.

The interviewees in the current study reported their inability to voluntarily reduce their use of the application, and recounted that they habitually use the application even when they drive on roads that they use every day. When the app is in use, it generates positive feelings for its users, such as calm and a sense of safety. Therefore, users' inability to reduce their use of the application is understandable.

The current study suggests that WAZE use does not resemble all behavioral aspects of technological addiction, including salience. The application does not dominate their thoughts

outside the context of driving. This finding is related to a new finding that emerges from the current study concerning practicality and functionality: WAZE, as a functional app, satisfies users' needs in specific circumstances (when driving), so users are not preoccupied with the application when it is not required. WAZE use does not come at the expense of other activities and is not characterized by users' immersion and loss of a sense of time during use, as the app is focused on the driving task and reaching the user's destination. This finding is consistent with findings of a previous study on the motives for WAZE use, which found that simplicity of use is a major motive [4].

Although some behavioral changes resulting from use of the application can be identified, these changes are specific and temporary. For example, the driver may reduce the car's speed in response to an alert from the app about the presence of police or a traffic camera, but the driver's behavioral response is dictated by the situation on the road to which the application the driver's attention, and not dictated by the application itself.

Furthermore, app users did not show evidence of tolerance, probably because the app fully satisfies users' need to reach their destination in the most efficient manner. The application is used for all drivers, and users' needs are fully satisfied by the application, which explains why tolerance does not increase as a result of extensive use of the application.

## Conclusions

Indeed, there are many ways to understand internet, mobile app, or social media use–including habits, or other behavioral mechanisms which don't require harm to be done for the use of an application to fit their definition, may be 'good' or 'bad' for users, or are more neutral than addiction [42–44]. Another might be reward-learning or conditioning, or work on deficient self-regulation [45]. The controversies that repeatedly swirl around world-wide platforms, such as the addictive appeal of some apps, reinforce the need for habit perspectives to clarify the relationship between social media and well-being. What is not controversial is the idea that social media use relies on habits. Bayer et al [43] for example, review of key studies and themes highlighted how habitual processes underpin a wide range of common uses and problems among users.

In our study, WAZE users exhibit behavioral patterns that are in line with the four symptoms of technological addiction–mood modification, conflict, relapse, and withdrawal–to varying degrees, and these patterns manifest as long as the application meets users' need to move efficiently from one location to another. The use of the application is framed by this need. Two additional symptoms of addiction–salience and tolerance–are not evident in WAZE use. However, functionality and satisfaction of the need to move efficiently from one location to another are manifested strongly.

The findings of the current study led to the conclusion that although the use of the application does not dominate WAZE users' thoughts out of the context of use, or require increasingly large doses of use to generate the same positive effect, WAZE users are highly dependent on the application in a way that may resemble technological dependency, which develops when an individual has no control over a technological source [1, 2].

At the same time, our specific framework should be mentioned as a potential limitation, given that the interviews are structured entirely around this framework and thus don't necessarily seek to answer questions that might have arisen from using another alternative framework mentioned above. Therefore, we conclude that even when application users exhibit only some of the symptoms that resemble dependency, it may indicate the existence of some degree of covert and quiet and undeclared dependency.

Covert dependency may increase as individuals increasingly use functional technological applications that effectively satisfy their needs, and are typically unconscious of their high

degree of reliance on these applications and their repetitive use. As a result of frequent use, users' functional needs are repeatedly satisfied in full, generating a sense of euphoria, serenity, and confidence, which explains why users choose to use the apps frequently, until such use conflicts an increasing number of other activities and competencies. When the user becomes totally dependent on these functional technological applications, and use is prevented, the user may experience withdrawal symptoms such as anxiety, fear, helplessness, and insecurity.

Practical implications may be drawn from the current research. The study indicates that functional applications have become beyond a utility that provides convenience and they have an absolute responsibility for the function itself. For this reason, the balance of power between the functional app designers and the users' needs to be reformed to a relationship of paid service provider and customer. The payment for the use of the application results in an obligation on the part of the application designed to maximize the quality of the application and its professionalism. In the current situation of free use of functional apps (freemium) the user is subject to a covert dependency of the services provided by the apps and therefor 'sells his soul', privacy and data to the free apps designers without the right or force to claim the most excellent services.

Future studies could examine whether the competencies that are in conflict with functional technological app use when users are in a state of dependency on the application may be redirected to other areas of the individual's life, such as leisure, enrichment, and empowerment activities. For example, individuals used to walk or run long distances to travel from one location to another. When various means of transportation obviated the need to do this, activities such as walking and running became part of leisure and sports activities.

## Research limitations

The interviewees' ages range from 23 to 55. Therefore, the research ignores an age group older than 55. Interviewing older interviewees may influence the conclusions, due to the significant technological divide in the baby boomer's generation. In addition, the research does not examine the association of demographic characteristics, such as, education, religion, economic status and marital status etc. to covert dependency.

## Author Contributions

**Conceptualization:** Tal Laor.

**Data curation:** Tal Laor, Yair Galily.

**Investigation:** Yair Galily.

**Methodology:** Yair Galily.

**Software:** Tal Laor.

**Supervision:** Yair Galily.

**Validation:** Yair Galily.

**Writing – original draft:** Tal Laor.

**Writing – review & editing:** Yair Galily.

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
