## [Decision Letter · Decision Letter 0]

29 Mar 2022

PONE-D-22-03742IN WAZE WE TRUST? GPS-BASED NAVIGATION APPLICATION USERS’ BEHAVIOR AND PATTERNS OF DEPENDENCYPLOS ONE

Dear Dr. Galily,

Thank you for submitting your manuscript to PLOS ONE. After careful consideration, we feel that it has merit but does not fully meet PLOS ONE’s publication criteria as it currently stands. Therefore, we invite you to submit a revised version of the manuscript that addresses the points raised during the review process.

We look forward to receiving your revised manuscript.

Kind regards,

Prabhat Mittal, Ph.D.

Academic Editor

PLOS ONE

Journal Requirements:

4. Please amend your authorship list in your manuscript file to include authors Yair Galily and Tal Laor. 

Additional Editor Comments:

The work reported in the paper possesses rigour and novelty. The authors may however extend the “Review of Literature” section. The authors may collate and compare these studies. In particular, the authors are advised to read and add the following papers:

“Opinion of students on online education during the COVID-19 pandemic”

“Designing Drawing Apps for Children: Artistic and Technological Factors”

“Role of emotion in addictive use of Twitter during COVID-19 imposed lockdown in India”

“Entrepreneurship education and employability skills: the mediating role of e-learning courses”

“Big data and analytics: a data management perspective in public administration”

“Control of COVID-19: A Counter Factual Analysis”

Reviewers' comments:

Reviewer's Responses to Questions

**Comments to the Author**

1. Is the manuscript technically sound, and do the data support the conclusions?

Reviewer #1: Yes

Reviewer #2: Yes

2. Has the statistical analysis been performed appropriately and rigorously? 

Reviewer #1: Yes

Reviewer #2: Yes

3. Have the authors made all data underlying the findings in their manuscript fully available?

Reviewer #1: Yes

Reviewer #2: Yes

4. Is the manuscript presented in an intelligible fashion and written in standard English?

Reviewer #1: Yes

Reviewer #2: Yes

5. Review Comments to the Author

Reviewer #1: Interesting and relevant article. The methodology is well described. However, it is missing the process of choosing the interviewees. How did the authors reach them? On-line? Personal acquaintance?

In my opinion, the discussion and conclusions section is detailed and concise. However, I would like the authors to emphasize the novelty of the current research. In addition, the authors conclude that users exhibit only several symptoms that resemble dependency. That is, even without dominating the user’s life (salience) or requiring ever-increasing frequency of use (tolerance), users of a functional application may exhibit high levels of mood modification, withdrawal, and relapse. As I found those findings very interesting, I would like to see the practical implications of it explained further.

It is assumed that every research has its own limitations and therefore I advise the authors to add the study limitations to the paper.

Reviewer #2: Very well done overall in the writing and literature review of this manuscript. The authors did a great job of framing the tech addiction to the research on Waze. There might be a benefit in adding the "novelty effect" of using such apps and tech for certain uses. In addition, it would be helpful to enhance the "limitations" portion of this paper to include all of the various permutations of what might have been missed throughout the 50 interviews that were conducted. Additionally it would be interesting to see the impact of new technology on the withdrawal symptoms and how the marketing efforts of app designer may impact the use of the various technologies and how their might be concerted efforts to get users to switch or to entice them with coupons, etc. Thank you for the opportunity to review this work, very timely and relevant.

6. PLOS authors have the option to publish the peer review history of their article (what does this mean?). If published, this will include your full peer review and any attached files.

Reviewer #1: **Yes: **Prof. Michael J. Leitner, Ph.D.

Reviewer #2: **Yes: **Simon Pack

---

## [Author Response · Author response to Decision Letter 0]

3 May 2022

Dear editor, 

Thank you and the reviewers for the feedback and for the careful consideration of our manuscript. We found the comments and suggestions to be thoughtful and valid. We have no doubt that the revised manuscript improved substantially because of the review process and our consequent revisions. Below, we point out the changes we have made to the manuscript vis-à-vis the comments, which are now integrated within the revised manuscript. 

Attached is the revised version of my manuscript " IN WAZE WE TRUST? GPS-BASED NAVIGATION APPLICATION USERS’ BEHAVIOR AND PATTERNS OF DEPENDENCY”. The additions are highlighted in yellow throughout the manuscript.

Once again, we would like to thank the reviewers and the editor for their encouragement and for the valuable suggestions for improvement. We have carefully considered all the remarks and suggestions listed in the review, resulting in a minor revision that, in our view, has clearly strengthened the paper:

Reviewer #1: 

Regarding comment 1: As suggested by the reviewer, we added to the methodology the entire process of choosing the interviewees. 

Regarding comment 2: As suggested by the reviewer, we emphasized the novelty of the research through the manuscript, highlighting the contribution to scholarship. The abstract has been revised to stress the essence of the novelty that lies in the current research. 

Regarding comment 3: As suggested by the reviewer, we added practical implications that may be drawn from the current research. Since the study indicates that functional applications have become beyond a utility that provides convenience and they have an absolute responsibility for the function itself we suggest that the balance of power between the functional app designers and the users needs to be reformed to a relationship of paid service provider and customer. 

Regarding comment 4: As suggested by the reviewer, we added the study limitations to the paper.

Reviewer #2: 

Regarding comment 1: As suggested by the reviewer, we added the "novelty effect" of using such apps and tech for certain uses including revising the abstract to emphasis the novelty that lies in the current research.

Regarding comment 2: As suggested by the reviewer, we enhanced the "limitations" portion of this paper to include all of the various permutations of what might have been missed throughout the 50 interviews that were conducted.

Regarding comment 3: As suggested by the reviewer, we added the impact of new technology on the withdrawal symptoms and how the marketing efforts of app designer may impact the use of the various technologies and the apps designers’ efforts to get users to switch or to entice them.

Best regards, 

Tal Laor and Yair Galily

---

## [Decision Letter · Decision Letter 1]

30 Aug 2022

PONE-D-22-03742R1IN WAZE WE TRUST? GPS-BASED NAVIGATION APPLICATION USERS’ BEHAVIOR AND PATTERNS OF DEPENDENCYPLOS ONE

Dear Dr. Galily,

we have received comments from a fourth reviewer for your submission. As per our previous discussions we leave the comments of reviewer 3 to your own discretion. The comments from Reviewer 4 are generally positive and highlight the need for more in depth discussion of the limitations. 

We look forward to receiving your revised manuscript.

Kind regards,

Carla Pegoraro

Staff Editor

PLOS ONE

Journal Requirements:

Additional Editor Comments (if provided):

Reviewers' comments:

Reviewer's Responses to Questions

**Comments to the Author**

1. If the authors have adequately addressed your comments raised in a previous round of review and you feel that this manuscript is now acceptable for publication, you may indicate that here to bypass the “Comments to the Author” section, enter your conflict of interest statement in the “Confidential to Editor” section, and submit your "Accept" recommendation.

Reviewer #1: All comments have been addressed

Reviewer #2: All comments have been addressed

Reviewer #3: (No Response)

Reviewer #4: All comments have been addressed

2. Is the manuscript technically sound, and do the data support the conclusions?

Reviewer #1: Yes

Reviewer #2: Yes

Reviewer #3: No

Reviewer #4: Yes

3. Has the statistical analysis been performed appropriately and rigorously? 

Reviewer #1: Yes

Reviewer #2: Yes

Reviewer #3: No

Reviewer #4: N/A

4. Have the authors made all data underlying the findings in their manuscript fully available?

Reviewer #1: Yes

Reviewer #2: Yes

Reviewer #3: No

Reviewer #4: Yes

5. Is the manuscript presented in an intelligible fashion and written in standard English?

Reviewer #1: Yes

Reviewer #2: Yes

Reviewer #3: Yes

Reviewer #4: Yes

6. Review Comments to the Author

Reviewer #1: All suggested revisions were made. The article is excellent and ready for publication. It is a very interesting and timely article.

Reviewer #2: (No Response)

Reviewer #3: The paper lacks the discussion about the aim of the study, the primary objective is not clear, significance and the comparison with the state of the art.

The introduction is quite verbose and does not cover the motivation and the contribution of the work. It needs proper extensions. It should state the motivation of the authors to conduct the present work and the way that it could be assistive to specific applications and systems. Background topics and related approaches are mainly illustrated in the Introduction. However, a better comparison with the author's approach should be made in the discussion or after the discussion section.

One of the limitations is the lack of discussion of the state of the art. The related work needs to be presented in better detail each one of the works addressed describing its aims (research questions specified) and the results collected.

The methodology section should better present the methodology. Currently, it is quite generic and verbose and needs to better present the exact stages of the methodology that it is followed and the functionality of each specific stage of it. Furthermore, you should make the research gap clear: Is it that, no one did before and why this would be interesting? Or is it the methodology? There is a major methodological issue to discuss: the relationship between Weibo input data, and how such data are correlated to the first scenic spot and then some given other spots. After all the discussion - what do we learn from your results? Who can profit from that? How your findings can be compared with findings from other authors. Please make a discussion that should be oriented towards the aims of the study and the gaps in the literature. Please highlight the innovation of the study too. Finally, the final patterns and trends discussed in the conclusion are essentially descriptive, the authors should here make every effort to discuss the social and/or economic reasons behind these patterns. One more important concern is the lack of discussion on the application of the current technique. Moreover, no compression what’s so ever has been made with the state of the art? The conclusion section needs to be re-written to better present the study. Finally, extensive editing of the English language and style required. The grammar needs polishing. Please have the paper proofread again. In the lights of the above-mentioned issues, the manuscript in the current form lack novelty, research contribution, and application. So it needs to be seriously rewritten and proofread

Reviewer #4: General Comments

Minor Revisions

This paper makes a valuable contribution to scholarship on the use of location-based smartphone apps, the use of a popular app like WAZE, and potentially similar apps including both google maps, apple maps, and other navigation or location-based apps. The qualitative interviews are well conducted and structured in ways aligned with prior qualitative research, as well as existing literature on technology dependence and internet addiction.

However, in using this framework without considering the implications or mapping it on to a relatively harmless act, the authors somewhat risk over-fitting regular/normal behavior into a pathological framework. This would likely be better served by being considered as a habit, routine, or impulse than an addiction. That may be something to consider for future studies, and the limitations of this should be acknowledged in the following ways:

The authors should add to the introduction (and/or discussion) that there are other ways to understand internet, mobile app, or social media use–including habits, or other behavioral mechaisms which don’t require harm to be done for the use of an application to fit their definition, may be ‘good’ or ‘bad’ for users, or are more neutral than addiction (LaRose, 2010; Bayer et al., 2022; Anderson & Wood, 2021). Another might be reward-learning or conditioning (as used in Lindstrom et al., 2021 to analyze Instagram data), or work on deficient self-regulation (Tokunaga, 2015). This should also include justification or reasoning for using the addiction framework for WAZE rather than one of these other known frameworks.

Further evidence of this issue is found in one part of the ‘dependency’ section, where anecdotes are used to justify saying that “Harm is reflected in developing a dependency on WAZE and a fear of leaving the home without it.” – this bordered on a very subjective interpretation of harm. One might, for example, also say that these users are being helped with their fear of driving by using this app. If not WAZE, these users would likely be dependent on another of the many navigation apps that exist. Absent a situation in which somehow, we lose all access to our smartphones (which, at the present point of society seems unlikely–and would not one simply pull over in the cases of phone overheat, or own car chargers to combat battery issues?) one might argue there is not any harm in being dependent on a maps app for navigation. If you are taking the view that depending on technology is an inherent bad or harmful, as the addiction framework seems to imply, then you reach (as these authors do here) towards interpreting this situation as harming the users. If you take another perspective you could say that this dependence is helping the users–or at least not harming them. In addition, this section provides no direct examples of harm to users or non-users related to this dependence (statistics of accidents, unsafe driving behavior, etc)--these might lend more support to this claim. I would recommend adding discussion of how the authors reasoned out their interpretation of the possible harms mentioned in this section in particular.

All of this considered, I think the authors do quite nicely in the conclusion to reject this potential pitfall of the framework used in the interviews, and rightfully note that not all of the qualities of technological/internet addiction apply in this case. Importantly, there are also legitimate harms mentioned in the “traces of relapse” section and mentions of unsafe and potentially harmful driving behavior due to WAZE dependence. Despite this, I think the use of this specific framework should be mentioned as a potential limitation, given that the interviews are structured entirely around this framework and thus don’t necessarily seek to answer questions that might have arisen from using another alternative framework (like the ones mentioned above).

With these minor revisions, I would recommend acceptance of the paper, which I think is quite strong and would be impactful for future research on this area.

Specific Notes

-Abstract has a sentence that is either missing a period or is incomplete…”as it meets users needs…”

-on PP 13 the sentence states that “In contrast to the younger group of interviews, the majority (20) of older interviewees stated that they would be reluctant to leave the house without WAZE” .. seems odd given the evidence presented is all suggesting that they would NOT be reluctant to leave the house, so I believe there’s a word missing here.

References:

Anderson, I. A., & Wood, W. (2021). Habits and the electronic herd: The psychology behind social media’s successes and failures. Consumer Psychology Review, 4(1), 83-99.

Bayer, J. B., Anderson, I. A., & Tokunaga, R. (2022). Building and breaking social media habits. Current Opinion in Psychology, 101303.

LaRose, R. (2010). The problem of media habits. Communication Theory, 20(2), 194-222.

Lindström, B., Bellander, M., Schultner, D. T., Chang, A., Tobler, P. N., & Amodio, D. M. (2021). A computational reward learning account of social media engagement. Nature communications, 12(1), 1-10.

Tokunaga, R. S. (2015). Perspectives on Internet addiction, problematic Internet use, and deficient self-regulation: Contributions of communication research. Annals of the International Communication A

7. PLOS authors have the option to publish the peer review history of their article (what does this mean?). If published, this will include your full peer review and any attached files.

Reviewer #1: **Yes: **Prof. Michael J. Leitner, Ph.D.

Reviewer #2: **Yes: **Simon M. Pack, Ph.D.

Reviewer #3: No

Reviewer #4: No

---

## [Author Response · Author response to Decision Letter 1]

3 Sep 2022

Thank you and the reviewers for the feedback and for the careful consideration of our manuscript, again (R2). We found the comments and suggestions to be thoughtful and valid. We have no doubt that the revised manuscript improved substantially because of the review process and our consequent revisions. Below, we point out the changes we have made to the manuscript vis-à-vis the comments, which are now integrated within the revised manuscript. 

Attached is the revised version of my manuscript " IN WAZE WE TRUST? GPS-BASED NAVIGATION APPLICATION USERS’ BEHAVIOR AND PATTERNS OF DEPENDENCY”. The additions are highlighted in yellow throughout the manuscript.

Once again, we would like to thank the reviewers and the editor for their encouragement and for the valuable suggestions for improvement. We have carefully considered all the remarks and suggestions listed in the review, resulting in a minor revision that, in our view, has clearly strengthened the paper:

WE have already answered Reviewers 1 & 2 in our previous version (R1);

*** PLEASE note that Reviewer 3 have review a different paper.. )-:

Reviewer 4: (There is a table in our cover letter which address ALL changes that have been made):

Comment Our Addition/Remark

This paper makes a valuable contribution to scholarship on the use of location-based smartphone apps, the use of a popular app like WAZE, and potentially similar apps including both google maps, apple maps, and other navigation or location-based apps. The qualitative interviews are well conducted and structured in ways aligned with prior qualitative research, as well as existing literature on technology dependence and internet addiction. MANY thanks. 

The authors should add to the introduction (and/or discussion) that there are other ways to understand internet, mobile app, or social media use–including habits, or other behavioral mechaisms which don’t require harm to be done for the use of an application to fit their definition, may be ‘good’ or ‘bad’ for users, or are more neutral than addiction (LaRose, 2010; Bayer et al., 2022; Anderson & Wood, 2021). Another might be reward-learning or conditioning (as used in Lindstrom et al., 2021 to analyze Instagram data), or work on deficient self-regulation (Tokunaga, 2015). This should also include justification or reasoning for using the addiction framework for WAZE rather than one of these other known frameworks. Indeed. We have added in the conclusion section the other ways to understand internet, mobile app, or social media use–including habits by using the work suggested (LaRose, 2010; Bayer et al., 2022; Anderson & Wood, 2021). And reward-learning or conditioning (as used in Lindstrom et al., 2021 to analyze Instagram data), or work on deficient self-regulation (Tokunaga, 2015).

All of this considered, I think the authors do quite nicely in the conclusion to reject this potential pitfall of the framework used in the interviews, and rightfully note that not all of the qualities of technological/internet addiction apply in this case. Importantly, there are also legitimate harms mentioned in the “traces of relapse” section and mentions of unsafe and potentially harmful driving behavior due to WAZE dependence. Despite this, I think the use of this specific framework should be mentioned as a potential limitation, given that the interviews are structured entirely around this framework and thus don’t necessarily seek to answer questions that might have arisen from using another alternative framework (like the ones mentioned above). Thank you. 

We have added in the text (conclusion) that “this specific framework should be mentioned as a potential limitation, given that the interviews are structured entirely around this framework and thus don’t necessarily seek to answer questions that might have arisen from using another alternative framework”.

With these minor revisions, I would recommend acceptance of the paper, which I think is quite strong and would be impactful for future research on this area. THANKS

Specific Notes

-Abstract has a sentence that is either missing a period or is incomplete…”as it meets users needs…”

-on PP 13 the sentence states that “In contrast to the younger group of interviews, the majority (20) of older interviewees stated that they would be reluctant to leave the house without WAZE” .. seems odd given the evidence presented is all suggesting that they would NOT be reluctant to leave the house, so I believe there’s a word missing here.

The abstract was corrected. 

Pp 13 also corrected. Not was added..

Best regards, 

Tal Laor and Yair Galily

---

## [Decision Letter · Decision Letter 2]

7 Oct 2022

IN WAZE WE TRUST? GPS-BASED NAVIGATION APPLICATION USERS’ BEHAVIOR AND PATTERNS OF DEPENDENCY

PONE-D-22-03742R2

Dear Dr. Galily,

We’re pleased to inform you that your manuscript has been judged scientifically suitable for publication and will be formally accepted for publication once it meets all outstanding technical requirements.

Kind regards,

Dr. Rizwan Muhammad

Guest Editor

PLOS ONE

Additional Editor Comments (optional):

Reviewers' comments:

Reviewer's Responses to Questions

**Comments to the Author**

1. If the authors have adequately addressed your comments raised in a previous round of review and you feel that this manuscript is now acceptable for publication, you may indicate that here to bypass the “Comments to the Author” section, enter your conflict of interest statement in the “Confidential to Editor” section, and submit your "Accept" recommendation.

Reviewer #1: All comments have been addressed

2. Is the manuscript technically sound, and do the data support the conclusions?

Reviewer #1: Yes

3. Has the statistical analysis been performed appropriately and rigorously? 

Reviewer #1: Yes

4. Have the authors made all data underlying the findings in their manuscript fully available?

Reviewer #1: Yes

5. Is the manuscript presented in an intelligible fashion and written in standard English?

Reviewer #1: Yes

6. Review Comments to the Author

Reviewer #1: Excellent article! Highly recommend that it be accepted for publication. The authors addressed all concerns from the prior reviews.

7. PLOS authors have the option to publish the peer review history of their article (what does this mean?). If published, this will include your full peer review and any attached files.

Reviewer #1: **Yes: **Prof. Michael J. Leitner

---

## [Editor Report · Acceptance letter]

13 Oct 2022

PONE-D-22-03742R2 

In Waze we trust? GPS-based navigation application users’ behavior and patterns of dependency 

Dear Dr. Galily:

I'm pleased to inform you that your manuscript has been deemed suitable for publication in PLOS ONE. Congratulations! Your manuscript is now with our production department. 

Kind regards, 

on behalf of

Dr. Muhammad Rizwan 

Guest Editor

PLOS ONE